# Visual Cortical Plasticity: Molecular Mechanisms as Revealed by Induction Paradigms in Rodents

**DOI:** 10.3390/ijms24054701

**Published:** 2023-02-28

**Authors:** Francisco M. Ribeiro, Miguel Castelo-Branco, Joana Gonçalves, João Martins

**Affiliations:** 1Coimbra Institute for Biomedical Imaging and Translational Research (CIBIT), University of Coimbra, 3000-548 Coimbra, Portugal; 2Institute for Nuclear Sciences Applied to Health (ICNAS), University of Coimbra, 3000-548 Coimbra, Portugal; 3Faculty of Medicine, University of Coimbra, 3000-548 Coimbra, Portugal

**Keywords:** visual cortex, ocular-dominance plasticity, stimulus-selective response potentiation, cross-modal plasticity

## Abstract

Assessing the molecular mechanism of synaptic plasticity in the cortex is vital for identifying potential targets in conditions marked by defective plasticity. In plasticity research, the visual cortex represents a target model for intense investigation, partly due to the availability of different in vivo plasticity-induction protocols. Here, we review two major protocols: ocular-dominance (OD) and cross-modal (CM) plasticity in rodents, highlighting the molecular signaling pathways involved. Each plasticity paradigm has also revealed the contribution of different populations of inhibitory and excitatory neurons at different time points. Since defective synaptic plasticity is common to various neurodevelopmental disorders, the potentially disrupted molecular and circuit alterations are discussed. Finally, new plasticity paradigms are presented, based on recent evidence. Stimulus-selective response potentiation (SRP) is one of the paradigms addressed. These options may provide answers to unsolved neurodevelopmental questions and offer tools to repair plasticity defects.

## 1. Introduction

The rodent visual cortex is an excellent neural circuit model for studying plasticity, as much of the detailed layer and interareal organization is well characterized. Several research tools have been applied to study anatomy and neural activity [1]. In addition, visual sensory inputs are easily accessible for manipulation. Several models have been developed and proposed to induce plasticity and elucidate the underlying molecular mechanism [2]. Hubel and Wiesel were the ones who described the first discovered paradigm: ocular dominance (OD) plasticity. In studies conducted in cats and monkeys, they observed that monocular deprivation (MD) leads to the shrinkage of the array of cells (OD columns) in the primary visual cortex (V1) related to the deprived eye and an expansion of the ones related to the spared eye [3,4]. Identifying the molecular pathways and the key molecules involved in plasticity is vital to understand conditions marked by defective synaptic plasticity, such as neurodevelopmental disorders [5]. Therefore, this review aims to characterize the molecular pathways involved in the two most studied plasticity paradigms: OD and cross-modal (CM) plasticity.

Signaling pathways are characterized both in healthy and in neurodevelopmental-disorder conditions. We refer to neural plasticity in broad terms, without strictly detailing the contributions of Hebbian vs. homeostatic plasticity and synaptic vs. intrinsic plasticity. Most of the studies address synaptic modifications; however, future studies may focus on the regulation of intrinsic excitability and hopefully fill this gap.

## 2. Manipulation of Plasticity in Healthy Conditions

### 2.1. Ocular Dominance Plasticity

The most-used paradigms to study the induction of plasticity in the V1 are the ones related to OD plasticity. This paradigm is induced after MD of one eye. The shift in the ratio between cortical responses to each of the two eyes is used to quantify this type of plasticity [2]. Moreover, there is a critical period (CP) where OD plasticity is higher, which in the mouse is between P21 and P35. Several cellular and molecular pathways are suggested to reach the mechanistic explanation for OD plasticity [6]. Hubel and Wiesel were the first who described this paradigm in kittens [3,4]. Since then, several lines of investigation have tried to uncover the molecular mechanisms responsible for OD plasticity and especially how to enhance it after the CP closure.

Catecholamines were the first proposed modulators of plasticity, and the first studies used intraventricular administration of a catecholamine toxin (6-hydroxydopamine (6-OHDA)) to totally deplete catecholamines from the cat brain. It was observed that in injected kittens there was a reduction in the OD shift to the unclosed eye, compared with vehicle-injected animals [7]. It is now known that the signaling pathway involves noradrenaline (NA), which binds to β-adrenergic receptors leading to cyclic adenosine monophosphate (cAMP) accumulation (Figure 1A). Protein kinase A will be activated and translocated to the nucleus-activating cAMP-responsive element-binding protein (CREB), which is known to regulate the expression of genes involved in neuronal plasticity and long-term-memory retrieval [8]. The activation of CREB involves its phosphorylation at three specific sites (Ser133, Ser142, and Ser143), as demonstrated in mouse and in vitro studies [9]. 

After MD, the response of the binocular V1 after stimulation of the non-deprived eye is enhanced, a process known to depend on *N*-methyl-d-aspartate receptor (NMDAR) [10]. Another target for OD plasticity during the CP that has been proposed is OTX2, which induces the transcription of Gadd45b/g and conducts the expression of plasticity-related genes, predominantly in parvalbumin-positive (PV) neurons in the adult visual cortex. An intriguing effect is observed in adulthood. Low levels of OTX2 result in Gadd45b/g upregulation (Figure 1G) [11]. Gadd45b/g is known for its role in long-term memory and plasticity [12]. 

The inhibitory activity of PV neurons has been associated with the regulation of CP for OD plasticity in the V1, although the molecular mechanisms still need to be fully clarified. A proposed mechanism is the neuregulin-1 (NER1)/ErbB4 signaling pathway, which has been shown to be downregulated after MD, leading to decreased inhibitory PV inputs onto excitatory neurons in the V1. This involves the retraction of excitatory inputs onto PV neurons, triggering the CP’s initiation (Figure 1F) [13,14]. Also acting on excitatory inputs onto PV neurons, neuropentraxin 2 (NPTX2) promotes the stabilization of α-amino-3-hydroxy-5-methyl-4-isoxazolepropionic acid receptor (AMPAR) in multiple cortical regions, including the visual cortex. Severin and colleagues performed an OD plasticity protocol in mice with MD. They observed that there is a reduction of connections between pyramidal and PV neurons in an early period, caused by reduced levels of NPTX2 [14] (Figure 1L). Both molecules are vital to allow OD plasticity through their disinhibitory role, thus enabling the mechanisms of Hebbian and homeostatic-synaptic plasticity to occur. 

Another signaling pathway well described for OD plasticity is the one triggered by the activation of Tropomyosin receptor kinase B (TrkB) [15]. By increasing TrkB phosphorylation in PV neurons, Winkel and colleagues were able to reopen the CP for OD plasticity in the adult V1 (P49-77) (Figure 1E). They performed the experiments using adult mice and observed increased OD plasticity after the administration of fluoxetine, known to bind TrkB. This shift was not observed in PV-specific conditional TrkB knockout mice [16]. 

Perineuronal nets (PNNs) in the adult cortex are crucial to enable structural integrity for neuronal circuits and adjacent cells [17]. It was shown that PNN digestion by chondroitinase ABC (chABC) was essential for TrkB signaling in PV neurons and for the consequent induction of OD plasticity [18]. ChABC was essential for degrading chondroitin sulfate proteoglycans (CSPGs) present in PNNs, which would otherwise bind to the protein tyrosine phosphatase σ (PTPσ), leading to TrkB inhibition (Figure 1E). TrkB signaling in PV neurons was shown to involve the modulation of voltage-gated potassium channels (Kv) 3 [16,19], contributing to decreased PV excitability, and enabling an increased plasticity status. Therefore, the mechanism described above may represent a second step in the cascade of mechanisms after the initial sudden disconnection of PV-pyramidal contacts. Besides the chABC, multiple enzymes exist in the PNN composition, particularly metalloproteinases (MMPs) responsible for ECM degradation (Figure 1). The administration of a specific inhibitor for MMP-2 and MMP-9 blocked plasticity in the non-deprived input using the OD paradigm in mice [17,20]. In addition, MMP-9^−/−^ mice exhibited impairments in OD plasticity after MD [21]. In PNNs, it was also observed that semaphorin3A, an important axon-guidance cue, is involved in OD plasticity [22]. After using specific tools to target semaphorin3, it was concluded that its accumulation in PNNs leads to the closure of CP for OD plasticity (Figure 1H) [22]. In addition, regarding PNNs, miR-29A was shown to be crucial for its stabilization, and, importantly, the inhibition of miR-29A was able to reduce the structural integrity of PNNs, leading to increased OD plasticity (Figure 1J) [23]. 

The study of OD plasticity focused mainly on understanding the molecular mechanisms involved in circuit rearrangement and altered excitation/inhibition balance. In this context, alterations in intrinsic neuronal properties have yet to be intensively studied. Rem2 is a Ras-like GTPase that plays an essential role in a signaling pathway necessary for plasticity (Ca^2+^-calmodulin). It was demonstrated in Rem2^−/−^ mice that this protein is vital for regulating cortical excitability in this sensory-dependent plasticity, as it prevents the hyperactivation of neurons, so that they can adjust their activity to the sensory inputs they receive (Figure 1C) [24].

A well-characterized signaling pathway involved in axon degeneration called Wallerian degeneration might also be relevant for OD plasticity. This pathway is known for the rapid degradation of axons following their separation from the cell body. This process depends on the reduction of cytoplasmic nicotinamide-mononucleotide adenyl transferases (NMNATs) [25]. In transgenic mice overexpressing NMNATs, it was observed that OD plasticity was compromised, though a clear mechanistic link between Wallerian degeneration with OD plasticity remains to be established (Figure 1D) [26]. 

Other cell types are also involved in OD plasticity. The research conducted in neuron maturation in the vertebrate visual system was extensive in exploring experience-dependent development. However, the contribution of glial cells is much less understood [27]. Microglia activity is being studied for its involvement in this type of plasticity. Indeed, microglial cells are known for their role in the physiology of synaptic plasticity [28]. Recently, a nonclassical molecule of the major histocompatibility class 1 (MHC1), Qa-1, was shown to be expressed in a group of excitatory neurons from cortical L6, and to play a vital role in OD-plasticity magnitude in the V1 [29]. The underlying mechanism was shown to depend on the microglial CD94-NKG2 receptor binding to Qa-1 [29]. It is known that microglial cells suffer alterations in their morphology, motility, or interaction with the synapse during plasticity induction [30]. Thus, future research should focus on detailing the intricated connections between different neuronal types and microglia. Another glial cell type with a predominant role in the CNS is the astrocyte. Although the role of astrocytes in the expression of plasticity is not fully elucidated, they have been implicated in CP OD plasticity. Adult mice transplanted with immature astrocytes showed restored OD plasticity [31]. Thus, the degree of astrocyte maturation is crucial for plasticity, and it is correlated with the transcriptional profile. Indeed, after the CP there is a tendency for a transition in the transcription of cell-division-related genes to cell-communication-related ones. Astrocytes produce a gap-junction protein connexin 30 that inhibits MMP-9, leading to a stabilization of PNNs whose structural integrity depends on PV neurons. The closure of the CP depends on the speed of maturation of this population of neurons (Figure 1K) [32]. Astrocytes also play an essential role in the reuptake of glutamate from the synaptic cleft, contributing to the regulation of neuronal excitability. Different excitatory amino-acid transporters are present in astrocytes [33], with glutamate transporter 1 (GLT1) being the principal player in glutamate uptake. Sipe and collaborators used a transgenic mouse line with constitutive heterozygous expression of the GLT1 (SLC1A2, GLT1-HET) and observed that MD in heterozygous mice lead to an aberrant OD plasticity in the ipsilateral non-deprived eye, characterized by an excessive depression of the responses (Figure 1M) [34]. An astrocytic protein called hevin was demonstrated to be vital to bridge pre-synaptic neurexin 1-alpha (NR1) and post-synaptic neuroligin 1B (NL1). Hevin is important for the recruitment of NL1 and NMDAR to excitatory synapses and, as result, the formation of thalamocortical synapses (Figure 1B). Hevin KO mice were exposed to 7 days of MD and revealed a deficient OD plasticity when compared with WT littermates [35]. Together, these results suggest that the astrocytic transcriptional profile and secreted proteins could impact plasticity processes both in the synapse and extracellular environment.

### 2.2. Cross-Modal Plasticity

When one sense is deprived, it is believed that remaining senses may show enhanced representations. This type of experience-dependent plasticity is called CM plasticity [36]. We can divide CM plasticity into two major groups: cross-modal recruitment, which uses the primary sensory cortex of the deprived sense for recruitment to the remaining senses; and compensatory plasticity, where there is a reorganization of the primary sensory cortex of the spared sense to allow for better sensory processing [37]. There are not many studies using CM-plasticity paradigms. Here, we report the existing evidence for molecular neuromodulation that might be involved in CM plasticity.

#### 2.2.1. Cross-Modal Recruitment

There is extensive evidence in humans that when one sensory modality is deprived, the respective cortical area is recruited for alternative sensory processing. The best example may be the blind human individuals that recruit the visual cortex when reading Braille [38].

At a circuit level, excitatory- and inhibitor-neurotransmission alterations have been observed in cross-modal plasticity-induction experiments with visual or auditory deprivation. The V1 is the most well-described cortical area for this type of plasticity, and will be detailed in this section. Animal vision loss can be induced by various methods, such as dark exposure, enucleation, intraocular-tetrodotoxin (TTX) injection, or lid-suture [39,40,41,42]. These methods are important for studying the circuit alterations in excitatory and inhibitory neurotransmission in V1 and its cross-modal recruitment from the remaining senses. After visual deprivation, many intracortical origins are described as responsible for the CM recruitment of the V1. Inputs that recruit L2/3 V1 arise from many intracortical origins, such as high-order visual areas (HVAs), sensory cortices, the prefrontal cortex, the retrosplenial cortex, or the V1. In addition, the V1 also receives inputs from subcortical areas, particularly the lateral posterior nucleus (LP), posterior thalamic nucleus (PO), and the lateral dorsal nucleus of the thalamus (LD) [37].

As with other neocortical brain regions, the V1 is organized in layers, with specific cell-types. The L4 is the thalamorecipient layer, also called the granular layer, due to the density of cells present. The superficial layers (L1 and L2/3) are called supragranular, whereas the deeper layers (L5 and L6) are called infragranular [43]. More superficial layers mature later than the L4. As a result, the CP for plasticity is more extended in superficial layers. [44]. This was observed in the cat visual cortex, where superficial layers had extended CP up to one year [44]. Therefore, L2/3 is more likely to develop plasticity in later periods than L4. In the V1 L2/3, there is an increased synaptic scaling after only two days of visual deprivation in mice, measured as increases in the amplitude of the miniature excitatory post-synaptic current (mEPSC). This effect is rapidly reversed after one day of light exposure, and this reversible change persisted after the CP for OD plasticity [45,46]. Additionally, Gao and colleagues found that a few days of visual deprivation decreased the frequency of miniature inhibitory post-synaptic currents (mIPSCs) in L2/3 pyramidal neurons of the V1. Together, the changes in mEPSC and mIPSC can be described as a form of homeostatic regulation in this cortical area [47,48]. 

The most accepted model is one in which visual deprivation leads to a reduction in overall responses in all layers of the V1, and, as a result, this lowers the synaptic-modification threshold for long-term potentiation (LTP), as shown for L2/3 [37,49]. The changes in the synaptic-modification threshold for LTP (or LTD) induction has been termed metaplasticity [37]. The final outcome is that previous subthreshold inputs, such as auditory information from the auditory cortex (A1), could be processed in the V1, particularly in L2/3.

One crucial feature of CM plasticity is the potentiation of lateral intracortical and subcortical inputs to L2/3 of the V1 after visual deprivation [50]. The inputs from the intact senses allow NMDAR-dependent plasticity [51]. Key neuromodulators have been identified in this plasticity paradigm, such as NA, acetylcholine (Ach), and serotonin (5-HT) (Figure 2A). Seol and colleagues conducted ex vivo studies in slices from 3-week-old rats to evaluate spike-timing-dependent plasticity. This was carried out in the visual cortex by stimulating L4 and recording responses from L2/3. With specific inhibitors of β-adrenergic or muscarinic receptors, they observed that the neuromodulators involved, particularly the activated downstream molecules (adenylate cyclase (AC) or phospholipase C (PLC)), could lead to LTP and long-term depression (LTD), respectively [52]. Therefore, the development of LTD or LTP depends not only on the order of timing, but also on the cholinergic and adrenergic neurotransmission. After these observations, Huang and collaborators concluded that receptors coupled to the Gs protein promote LTP and suppress LTD, whereas receptors coupled to the Gq11 protein promote LTD and suppress LTP. In rat (P20-30) brain slices, responses were recorded from L4 and L2/3 and the appearance of the LTP- and LTD-only states was observed. In vivo studies confirmed the same observations [53]. Both ACh and NA are involved in the arousal status [54]. Therefore, the alertness status of the animal may impact plasticity induction. 5-HT and ACh act on the VIP neurons at the circuit level in the V1. Particularly, 5-HT acts on 5-HT-3A receptors of the VIP neurons. By performing a monocular-enucleation paradigm to induce unilateral visual loss, molecular analysis revealed that this class of receptor in the VIP neurons was vital for the late-phase recruitment of monocular visual cortex for whisker-mediated sensory processing [55]. The VIP neurons carry cross-modal information from the other sensory cortices projecting to L2/3 of the V1 [56,57]. By inhibiting the somatostatin-positive (SST) and PV neurons, they play a disinhibitory role, enabling an increased excitatory activity of pyramidal neurons in the deprived cortex [58,59,60].

#### 2.2.2. Cross-Modal Compensatory Plasticity

The CM compensatory plasticity occurs when the cortical areas for the spared senses are reorganized to allow better processing of their own sensory information [36]. Although recent evidence suggests compensatory plasticity occurs in humans [61], most of the molecular insight comes from animal studies [62]. For instance, in adult mice with whisker deprivation, it was verified that V1-dependent vision received a considerable boost, and increased visual acuity and contrast sensitivity were observed at a behavioral level [63]. 

The proposed neuromodulatory basis for CM compensatory plasticity depends on the neurotransmission mediated by 5-HT, NA, and Ach (Figure 2B). In the experimental paradigms for inducing this type of plasticity with visual deprivation, increases in the levels of NA, 5-HT and dopamine (DA) in the spared cortices have been found [64,65]. The effects of neuromodulation in a type of compensatory-plasticity paradigm were firstly addressed by Qu and colleagues, in experiments with cats with partial retinal lesion. They observed that the concentration of NA, 5-HT, and DA in the spared visual-cortical regions was higher than in the deprived cortex or the cortex of control animals [65]. In agreement with the results in cats, Jitsuki and collaborators showed that after visual deprivation in rats, the barrel cortex presented elevated levels of 5-HT [64]. The increased levels of this neuromodulator lead to the strengthening of the L2/3 synapses in the barrel cortex and to increased feedforward processing of whisker information. They were able to unveil the molecular mechanisms involved in this potentiation that involve the activation of the ERK pathway and the downstream phosphorylation of the GluR1 AMPAR subunit. AMPAR will travel into these synapses to increase strength and connectivity [64]. The elevated levels of ACh and NA are related to the elevation of arousal and increased attention [66,67,68]. Therefore, the behavioral state of the animal is a determinant of the development of this form of plasticity. In the spared cortices, ACh acts through two distinct types of receptors: nicotinic and muscarinic. Nicotinic-receptor activation leads to the potentiation of the feedforward thalamocortical synapses, whereas muscarinic receptors depress the lateral intracortical and thalamocortical inputs [69]. This was observed in L3 excitatory post-synaptic potentials (EPSPs) in rat and mouse brain slices. NA activates β-adrenergic receptors, which facilitates the induction of LTP and the suppression of LTD, as was mentioned in cross-modal recruitment. The elevated levels of noradrenergic transmission induced after visual deprivation facilitates the potentiation of the feedforward synapse in the spared cortex. (Figure 2B) [64].

## 3. Neurodevelopment-Disorder Conditions

Studies of cortical plasticity started with Hubel and Wiesel, focusing on a healthy condition during normal neurodevelopment, particularly the V1. They observed an innate development period and another CP for experience-dependent plasticity [3,4]. What happens in these CPs is very important for normal neurodevelopment, which is why this is a crucial step, deserving comprehensive investigation within neurodevelopmental disorders [2,70].

### Ocular-Dominance Plasticity

The visual system is affected in neurodevelopmental disorders such as autism spectrum disorders (ASD). For instance, binocular rivalry, a process characterized by alternating visual perception between the two eyes, has a slower alternation rate in the autistic brain. A deficient excitation/inhibition balance contributes to this altered plasticity in the visual cortex [71,72]. 

Plasticity defects may be seen in conditions marked by impaired neurodevelopment. In a mouse model of neurofibromatosis type 1, a model to study ASD, high levels of inhibition were observed in the adult V1, with no overt effects on early cortical development [73]. The authors found increased inhibition during development to be the cause of early closure of the CP for OD plasticity. Shank3 mutant mice present autism-like behaviors and have been also used as an animal model for ASD [74]. Using Shank3 mutant mice, Tatavarty and colleagues observed a disruption in OD plasticity measured by the OD index, and suggested a role for Shank3 in homeostatic compensatory mechanisms or synaptic scaling [75]. Ube3, a ubiquitin ligase involved in autism and Angelman syndrome, also impacted OD plasticity, as confirmed with in vivo studies [76]. When a region of chromosome 16p11.2 is micro-deleted, there is an increased susceptibility for ASD and one candidate gene in this region is the major vault protein *(MVP)*. Ip and collaborators found that MVP^+/−^ mice presented reduced OD plasticity after MD, and suggested that the possible molecular pathways which are disrupted depend on STAT1 and ERK signaling [77]. Future works are needed to fully clarify all the molecular pathways in genetic models of ASD and other neurodevelopmental conditions. With a complete picture of the molecular cascade disrupted in ASD, targeted therapeutic strategies could be addressed. 

## 4. Other Plasticity Paradigms

Stimulus-selective response potentiation (SRP) is a plasticity paradigm that consists of visual-response potentiation by repeated exposure to a particular visual-stimulus orientation [78]. This protocol enables discrimination between a familiar and a novel stimulus. Electrophysiological recordings and behavioral analysis identified the fact that orientation-selective habituation to a visual stimulus (OSH) requires storing information in the V1 [79]. SRP is known to require NMDAR in the V1, although there are distinct laminar requirements. Indeed, Fong and colleagues found that, in L4, NMDARs are only negligibly involved [80]. SRP relies on PV neuron-activity modulation. Khan and collaborators evaluated learning-induced changes in stimulus selectivity and interactions in the GABAergic interneurons in the V1, and, notably, it was seen that PV interneurons are as selective as pyramidal cells, and that with learning, they organize stimulus-selective pyramidal-PV ensembles. This effect was not observed in other classes of interneurons, such as SST and VIP [81]. Montgomery and colleagues recently reviewed some of the already well-established hypotheses and potential circuit models, to explain SRP [82]. It is crucial to detail not only the intricate circuit organization but also a complete mapping of the molecules and signaling pathways involved. With that information, we can identify upstream targets amenable to modulation.

The previously mentioned plasticity paradigms are the most well-described, and they can provide answers about some aspects of synaptic plasticity and its main molecular mechanisms. However, in order to understand how complex environmental information contributes to such plasticity, we must add contextual complexity to the paradigms. Indeed, some evidence has been provided about the influence of the environmental cues in the V1. Therefore, plasticity in the V1 depends on these cues, and they are worth exploring [83,84,85,86,87]. Indeed, the visual system is the foremost option for studying synaptic plasticity, since it is a sensory cortical region that is easy to stimulate and to develop experimental plasticity-induction protocols. The establishment of plasticity in the V1 depends on environmental cues, particularly navigation signals. These navigational signals include speed, distance traveled, or head movement [83,88]. In addition, increased locomotion activity leads to a consequent increase in the arousal status. This increased arousal leads to increased activity of visually evoked neurons in the V1 [85]. In a recent study, while mice explored a virtual-reality corridor with two identical landmarks, it was observed that CA1 and V1 neurons encoded the animal position in the corridor [84]. A study conducted in awake mice using two-photon calcium imaging revealed that the representation of orientation and direction of drifting gratings in the V1 was improved in an audiovisual context (multimodal integration), rather than in a visual context (unimodal integration) [87]. This study contributes to a better understanding of cross-modal sensory integration and cross-modal plasticity. In addition, some conditions of the environment, such as food scarcity, can also modulate visual responses in mice [86]. The fat-mass-regulated hormone leptin mediates this effect. In mice, food restriction was shown to decrease ATP usage by mouse V1 neurons. The authors also observed reduced AMPAR currents in the V1 and a consequent reduction in visual precision, manifested by a broadened orientation tuning [86]. 

The optimization of in vivo plasticity protocols based on the evidence from these studies would allow a wider understanding of the mechanisms involved, and, more importantly, would unveil the key molecular players responsible for sculpting neural circuits during development.

## 5. Conclusions and Future Perspectives

The use of plasticity-induction paradigms in vivo has contributed to revealing multiple molecular players involved in brain maturation. However, many observed processes remain poorly characterized and deserve further investigation. Induced plasticity paradigms thus represent opportunities to deepen this knowledge and possibly find previously unknown players in brain maturation.

Although most studies have focused on synaptic modifications, we cannot discard the involvement of intrinsic-excitability modifications (intrinsic plasticity). Some studies that used visual-deprivation protocols have identified the involvement of ion-channel modulation as part of intrinsic-plasticity mechanisms [89,90]. However, there are very few studies addressing in vivo intrinsic neuronal properties [91,92]. The majority of the studies were performed using in vitro models, as reviewed by Debanne and collaborators [93]. Therefore, the development of appropriate induction protocols and the study of intrinsic excitability should be considered in future studies.

It is during the CP for OD plasticity that binocularity and binocular matching develop [2,7,42]. The complete identification of the molecular mechanisms underlying the opening and closure of the CP will provide researchers with tools to enhance or correct plasticity when it may be compromised, such as in ageing and neurodevelopmental disorders [5,94,95]. As exemplified in Figure 1, some of the possible interventions may address the glutamate-receptor or TrkB signaling pathways or PNN stability. The latter has emerged as a converging mechanism in many of the pathways identified [96].

Environmental complexity may also be relevant to understand plasticity. Accordingly, alterations in gut microbiota were shown to mediate visual-cortical plasticity in an environmental-enrichment context [97]. Gut microbiota dysbiosis has been observed in ASD, and the cross-talk between the brain and intestinal microbiota contributes to the phenotypic manifestations of ASD [98,99]. Therefore, OD plasticity might be compromised in ASD, and the molecular mechanisms disrupted. It could be interesting to address OD in ASD with microbiota manipulations, to investigate specific molecular pathways that could be therapeutically targeted.

The initial cortical developmental window is characterized by cross-modal plasticity, which is vital for developing sensory cortices and may be disrupted in neurodevelopmental disorders such as ASD. Interventions aimed at restoring the correct sensory development during this initial cross-modal stage may be useful to ameliorate later impairments. As an example, oxytocin is an essential neuromodulator and is downregulated in neurodevelopmental conditions [100,101]. Indeed, different pre-clinical and clinical studies have evaluated the impact of the administration of oxytocin on rescuing some of the social features presented in children with ASD [102,103,104]. Interestingly, Zheng and colleagues found that in neonatal mice, oxytocin administration elevated excitatory-synaptic transmission in multiple sensory cortices and, consequently, they observed an increased sensory experience. In a condition where one sensory modality is deprived, both environmental enrichment and oxytocin could be interesting for increasing excitatory-synaptic transmission in the deprived cortex, in a cross-modal recruitment process [105]. Since ASD patients are characterized by hypersensitivity or hyposensitivity [106], oxytocin may be investigated, due to its role in this therapeutic window in ASD to normalize sensory experience.

Finally, the defective maturation of fast-spiking PV neurons is being pointed out as a common mechanism for neurodevelopmental conditions (autism, schizophrenia, and bipolar disorders) [107]. The deletion of neurofibromin 1 (NF1) from GABAergic interneurons was shown to impact PV neuron maturation [108]. This deletion leads to the downregulation of the Lh6 transcription factor, which is fundamental for the maturation of PV neurons. The signaling pathway NF1/Ras/MEK regulates the maturation of the PV neurons, and they are the primary source of circuit inhibition, and are vital for SRP development [82]. SRP is an interesting paradigm in neuropsychiatric conditions characterized by defective neurodevelopment. VIP neurons play a crucial role in plasticity development in cross-modal recruitment, whereas PV neurons are essential players in SRP. Therefore, this demonstrates the fact that no single population of cells is a potential common target for the different plasticity paradigms.

In conclusion, the previously detailed synaptic-plasticity paradigms are the best understood, as documented in the cited literature. A complete understanding of the molecular mechanisms involved in each paradigm is needed for healthy conditions and neurodevelopmental disorders. A detailed picture of the molecular pathways involved will provide researchers with better tools to amend the conditions of defective plasticity while identifying novel therapeutic targets. 

## Figures and Tables

**Figure 1 ijms-24-04701-f001:**
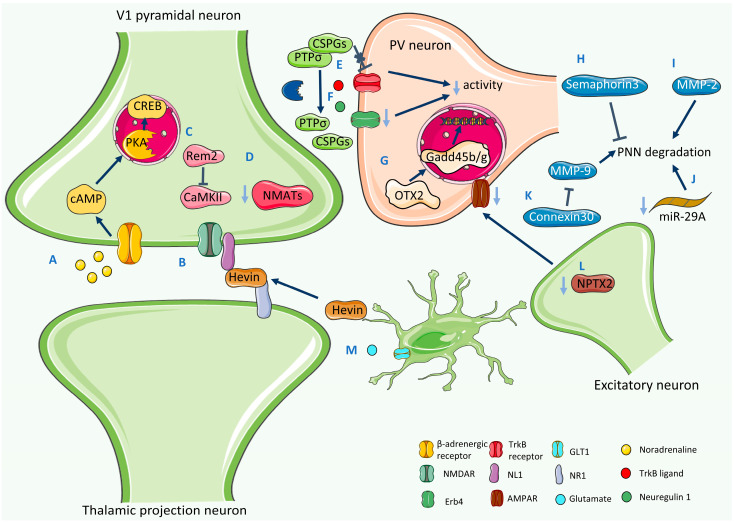
Molecular characterization of ocular-dominance (OD) plasticity in healthy conditions. (**A**) Noradrenaline (NA) targets β-adrenergic receptor causing cyclic adenosine monophosphate (cAMP) increase and protein kinase A (PKA) activation. PKA is then translocated to the nucleus and activates the cAMP-responsive element-binding protein (CREB) transcription factor responsible for the regulation of genes involved in synaptic plasticity. (**B**) Astrocytic hevin bridges neurexin 1-alpha (NR1) of thalamic pre-synaptic terminals with neurolignin 1B (NL1) and the *N*-methyl-d-aspartate receptor (NMDAR) cortical post-synaptic terminals. (**C**) Intracellularly, Rem2 is important for inhibiting Ca^2+^-calmodulin-dependent protein kinase II (CaMKII) to regulate cortical excitability. (**D**) Cytoplasmic nicotinamide-mononucleotide adenyl transferases (NMATS) might be reduced in OD plasticity. (**E**) In perineuronal nets (PNNs), chondroitinase ABC (ChABC) is an enzyme that degrades and cleaves protein tyrosine phosphatase σ (PTPσ) and chondroitin sulfate proteoglycans (CSPGS). This prevents PTPσ from inhibiting Tropomyosin receptor kinase B (TrkB) in parvalbumin-positive (PV) neurons, leading to a decrease in the excitability of these neurons. (**F**) Downregulation of the neuregulin-1 (NER1)/ErbB4 pathway is important to reduce inhibitory inputs during the critical period. (**G**) Predominantly in PV neurons, OTX2 induces the transcription of Gadd45b/g which is associated with the expression of plasticity-related genes. (**H**) Elevated levels of semaphorin3 stabilize PNNs and close critical period (CP) for OD plasticity. (**I**) Metalloproteinase 9 (MMP-9) and metalloproteinase 2 (MMP-2) activity are important for PNN degradation, which is important for OD-plasticity manifestation. (**J**) The reduction in miR-29A destabilizes PNNs and promotes plasticity. (**K**) Astrocyte (green) releases connexin30, which inhibits MMP-9 and stabilizes which PNNs, that impact OD-plasticity development. (**L**) A decrease in neuropentraxin 2 (NPTX2) levels is associated with a reduction in the excitatory inputs onto the PV neurons, leading to increased plasticity. (**M**) Astrocytic-glutamate reuptake via glutamate transporter 1 (GLT1) is crucial for OD-plasticity responses in the non-deprived eye. Dark blue arrows indicate activation and T-shaped lines indicate inhibition. T-shaped line with X indicates disinhibition. Light blue vertical down arrows indicate downregulation. This figure was partly generated using Servier Medical Art by Servier, licensed under a Creative Commons Attribution 3.0 unported license.

**Figure 2 ijms-24-04701-f002:**
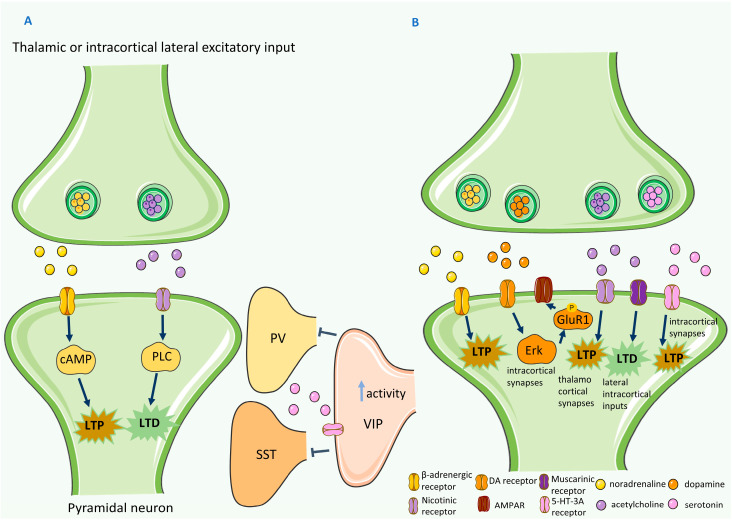
Cross-modal (CM) plasticity molecular mechanisms in a healthy condition: (**A**) CM recruitment. Noradrenaline (NA) will activate β-adrenergic receptors, and cyclic adenosine monophosphate (cAMP) will be activated downstream, inducing long-term potentiation (LTP). Acetylcholine (Ach) targets the muscarinic receptors, L/3 V1 pyramidal neurons, activating phospholipase C and promoting LTD. Serotonin (5-HT) acts on 5-HT-3A receptors in the vasoactive intestinal peptide-positive (VIP) neurons and inhibits parvalbumin-positive (PV) and somatostatin-positive (SST) neurons. (**B**) CM compensatory plasticity. NA and Ach are both involved in LTP induction in the spared cortex. ACh acts through nicotinic and muscarinic receptors, promoting the potentiation of thalamocortical feedforward synapses and depressing lateral intracortical inputs. Dopamine (DA) will activate the ERK signaling pathway that will phosphorylate downstream glutamate receptor 1 (GluR1), the amino-3-hydroxy-5-methyl-4-isoxazolepropionic acid receptor (AMPAR) subunit, promoting the travel of AMPAR to the synapse to strengthen the connections. 5-HT will promote the development of LTP in the spared cortex. Dark blue arrows indicate activation and T-shaped lines indicate inhibition. Light blue vertical up arrow indicates increased activity. This figure was partly generated using Servier Medical Art by Servier, licensed under a Creative Commons Attribution 3.0 unported license.

## Data Availability

Not applicable.

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
