# Peer review of "Visual Cortical Plasticity: Molecular Mechanisms as Revealed by Induction Paradigms in Rodents"

_ijms, 2023, doi:10.3390/ijms24054701_

Round 1

Author Response

Reviewer report 1:

  1. “When referring to the visual cortex, please specify if the primary visual cortex

or high visual areas are included."

Thank you for noticing this, we have now detailed “primary visual cortex” or “V1” whenever it was not clear if other visual areas were included.

  1. “Move the discussion in lines 44-47 to the conclusions section and expand on

the described mechanisms for opening and closing the CP and potential ways

of expanding or reopening it. Add missing citation in line 48.”

As suggested, we have moved that information segment from topic 2.1 to topic 5, and expand on mechanisms, and add missing citation.

  1. “For readers new to the topic, provide a brief overview of the main findings

encountered by Hubel and Wiesel in the experiments described in lines 310-

313 in the introduction or conclusions section.”

As suggested, we have included some information about the main findings of Hubel and Wiesel in topic 1, the introduction section.

  1. “Be more specific about PV. It needs to be clear if the mechanisms described in

line 72 refer only to the visual cortex.”

We have clarified that we were referring to ocular dominance plasticity and primary visual cortex V1.

  1. In line 138, please clarify if the sentence refers to OD plasticity or if it is in line

with astrocytes' general contribution to plasticity. Review this paragraph and

include references such as Wahis et al. Current Op. Neuro 2021 and Benfey et

al Frontiers in Neuron Circuits 2022.

We clarified the sentence suggested and added two sentences to better chain the ideas in the paragraph with the Benfey et al. Frontiers in Neuron Circuits 2022 reference as it fits better the review's aim.

  1. “Missing citations in line 188”

Reference added in topic 2.2.

  1. “In line 228, rephrase and add missing references. Clearly define the responses

to stimuli in V1 or other areas and explain the concepts of plasticity and

metaplasticity used in the review.”

We have changed this and previous sentences trying to better explain the experiments and conclusion. In addition, we clarify the question about Hebbian vs homeostatic plasticity and synaptic vs intrinsic plasticity in the introduction.

  1. “The paragraph about cross-modal compensatory plasticity may be misleading.

Please revise and ensure that it aligns with the main ideas the authors aim to

convey.”

Thank you for noting this, we have revised the paragraph attempting to make it more clear.

  1. “Consider adding a scheme or figure to describe how the plasticity protocols

discussed in detail are induced in each case.”

We agree a scheme may be useful to illustrate the indicated protocols, however, there is no coherence among different studies, either in the type of visual deprivation or the animal age or the time of deprivation. We think it may be confusing as we detail these aspects. However, we will be willing to add a scheme, figure, or table if the reviewer consider it is important.

  1. “Paragraph 396 appears disconnected from the rest of the review. Please

reorganize and determine if it should be included in the review.”

We think that this paragraph is interesting since it refers to possible therapeutic implications of the knowledge provided by the use of plasticity paradigms, especially in the context of neurodevelopmental disorders. We have rephrased the paragraph attempting to make it more clear. However, we agree it is not a mandatory topic, and we will be willing to remove it if the reviewer consider it is confusing/disconnected with the rest of review.

Reviewer 2 Report

This is an interesting review paper aimed at reviewing the molecular mechanisms of visual cortical plasticity. The paper is well written, interesting and timely. However, I have a few comments that should be addressed.

Specific comments:

1. The plasticity involving ion channel regulation (i.e., intrinsic plasticity) has surprisingly not been really considered in this review, despite evidence in the field of functional plasticity in the visual cortex (Maffei & Turrigiano, J Neurosci 2008; Nataraj et al Neuron 2010; Debanne et al., Curr Opin Neurobio,…). I suggest the authors to include a chapter on this purpose.

Minor comments:

1. Please revise the title of Figure 1, not clear as it is.

2. Figure 1 is a bit messy. Please, consider to clarify this figure.

Author Response

Reviewer report 2:

  1. “The plasticity involving ion channel regulation (i.e., intrinsic plasticity) has surprisingly not been really considered in this review, despite evidence in the field of functional plasticity in the visual cortex (Maffei & Turrigiano, J Neurosci 2008; Nataraj et al Neuron2010; Debanne et al., Curr Opin Neurobio,…). I suggest the authors to include a chapter on this purpose.”

We really thank the reviewer for raising this issue. We have unintentionally omitted clear references to either Hebbian vs homeostatic plasticity and also omit discussion on the contribution of synaptic vs intrinsic plasticity. However, the purpose of the review is to exemplify the value of using plasticity induction paradigms, in particular in vivo protocols and behaviorally relevant, as they have revealed important molecular players. We have clarified in the manuscript that the intention of the review is to indicate molecular players of more broadly functional neural plasticity (related to mentioned paradigms only) without strictly detailing the involvement of Hebbian vs homeostatic plasticity or synaptic vs intrinsic plasticity. Most of studies focus on synaptic modifications but we cannot discard the contribution of intrinsic excitability modifications. However, there are very few studies specifically addressing this issue in vivo, especially in the context of the mentioned paradigms.

In order to meet the reviewer suggestion, we have also included a paragraph in the conclusion addressing intrinsic excitability.

  1. “Please revise the title of Figure 1, not clear as it is.”

The title of the figure has been altered to “Molecular characterization of ocular dominance plasticity (OD) in healthy conditions.

  1. “Figure 1 is a bit messy. Please, consider to clarify this figure.”

We agree the figure 1 is a bit messy. We have removed the microglia cell since there was not a direct link to neurons, and then re-arranged the rest of figure hoping it is now more clear.
